# Psychological Distress among Caregivers of Children with Neurodevelopmental Disorders in Nepal

**DOI:** 10.3390/ijerph18052460

**Published:** 2021-03-02

**Authors:** Hans Kristian Maridal, Hanne Marit Bjørgaas, Kristen Hagen, Egil Jonsbu, Pashupati Mahat, Shankar Malakar, Signe Dørheim

**Affiliations:** 1Department of Psychiatry, Molde Hospital, Møre and Romsdal Hospital Trust, 6412 Molde, Norway; kristen.hagen@helse-mr.no (K.H.); egil.jonsbu@helse-mr.no (E.J.); 2Department of Research, Division of Psychiatry, Haukeland University Hospital, Health Bergen, 5021 Bergen, Norway; 3Department of Pediatric Neurology/Habilitation, Habu, Stavanger University Hospital, 4068 Stavanger, Norway; hanne.marit.bjorgaas@sus.no; 4Department of Mental Health, Faculty of Medicine and Health Science, NTNU, 7491 Trondheim, Norway; 5Centre for Mental Health and Counseling Nepal, Thapathali, Jeetjung Marg, Kathmandu 44600, Nepal; drpashupati@mos.com.np (P.M.); shankar.malakar@gmail.com (S.M.); 6Sandnes District Psychiatric Centre, Division of Psychiatry, Stavanger University Hospital, 4068 Stavanger, Norway; signe.karen.dorheim@sus.no

**Keywords:** neurodevelopmental disorder, disability, caregiver, psychological distress, low-income country, depression, anxiety, burden, Nepal

## Abstract

Parenting a child with neurodevelopmental disorder (NDD) is related to a higher rate of anxiety and depression, increased stress, and reduced quality of life. Although there is reason to believe that parenting children with NDD in low- and middle-income countries (LMIC) can be challenging, there is a lack of knowledge on the psychological distress among these caregivers, especially in rural areas. The aim of the study was to examine the psychological distress among caregivers having children with NDD in rural Nepal. Sixty-three caregivers were visited in their homes and interviewed by experienced mental health professionals. This study examined demographic information, severity of disability, perceived caregiver burden, and psychological distress, measured by the General Health Questionnaire-12 (GHQ-12). The study found a high level of psychological distress in the caregivers (*M* = 5.38, *SD* = 2.8). A majority (90.5%) scored two or higher, indicating the presence of a common mental disorder (CMD). Almost half (46%) scored six or higher, indicating a high level of distress. A majority of the caregivers reported that caring for their disabled child had a negative effect on the caregiver’s economy (70%), physical health (65%), social life (64%), and dreams and expectations for the future (81%). There was a significant relationship between the caregiver’s psychological distress (GHQ-12) and degree of disability in the child (Gross Motor Function Classification System), degree of caregiver burden, feeding problems, having health workers as a possible source of help, receiving incentive from the government, having somebody to confide in, and caregiver illiteracy. A forward regression analysis entering the significant factors indicated that caregiver burden, having someone to confide in, and having health workers as a possible source of help were significant related to psychological distress. The final step of the model explained 42.4% of the variance in psychological distress among the caregivers. The study indicates a high level of psychological distress and high overall burden in caregivers of children with NDD in rural Nepal. Further implications for research and service development are discussed.

## 1. Background

Parenting a child with neurodevelopmental disorder (NDD) can be challenging [1,2]. In some parts of the world this challenge can be further exacerbated by poverty and lack of support from the surroundings [3]. NDD is a term used for a range of disorders that involve injuries to the developing brain, such as cerebral palsy and genetic disorders, and also intellectual disability and other conditions that lead to functional limitations such as impaired cognition, motor performance, vision, hearing, and speech, and behavior problems [4,5]. There is a lack of epidemiological data on NDD prevalence in low- and middle-income countries (LMIC) [6], but it is estimated that between 93 and 150 million children suffer from some kind of disability, and most of these live in low-income countries [7,8]. Children with disabilities are often not represented in official statistics and as a result remain politically and socially “invisible” [9]. Overall, NDD conditions are likely to be more common in low- and middle-income countries (LMIC) because of the higher prevalence of recognized risk factors such as poor maternal and child healthcare in the prenatal and postnatal period, nutritional deficiencies, and neonatal infections [6].

NDD is a public health challenge for LMIC [10] on account of their high prevalence, lifetime duration, and considerable social, personal, and economic consequences, as well as the impact on educational and employment opportunities in later life. Treatment, prevention, and management of these conditions are often not considered in healthcare plans, although they place a heavy burden on healthcare systems [11].

There is a growing understanding of the impact of NDD on the quality of life and mental health of families with children with these conditions. However, the majority of research has been conducted in high-income countries, and there is a lack of knowledge on the situation in low- and middle-income countries [12]. Research from high-income countries has shown substantially increased stress [1,2,13,14], chronicity of distress [15], fatigue [2], higher rate of anxiety and depression [12,16], emotional and cognitive problems [17], and a greater likelihood of having physical illness [17,18] for these caregivers. They also forfeit employment opportunities, leisure time, and social interactions and often become housebound [2].

Moreover, there are also substantial economic challenges in having a child with a disability [19]. In LMIC with little or no governmental support, this is believed to be an even greater problem and can make caregivers impoverished and indebted [20].

Caregiving experiences in different contexts may also differ, as there are many additional challenges in raising a child with NDD in LMIC [3,21]. Social isolation and psychological distress can be further exacerbated by stigma, exclusion, and inequitable policies [2]. 

The presence of psychiatric distress in the caregiver is associated with reports of poorer social support, family dysfunction, greater adverse impact of the child’s situation on the family, poorer child behavior, unfavorable parenting styles, poorer child psychosocial functioning [22], and even a negative impact on siblings and marriage [23]. Maternal depression has been shown to affect the child’s psychological, intellectual, and psychosocial development, and may increase the risk of malnourishment and physical as well as psychiatric illness in the child [4]. The resources, family support, and overall functioning of the family are important factors for achieving a satisfying life for children with disabilities [24]. It is therefore important to gain more knowledge about the psychological distress of these parents in LMIC.

According to Thrush and Hyder, the lack of information, especially quantitative data, on the situation of caregivers and their burden in LMIC allows the perpetuation of the misleading notion that families in LMIC are capable of providing care without public or private institutional support [3], and better evidence would provide incentive for governments of LMIC to put the needs of caregivers higher on their agenda.

Some studies have shown that caregivers in low-income countries seem to have a higher risk of suffering from some form of psychiatric morbidity [25,26,27,28,29]. Disability in the child may have a large impact on the lives, economy, and mental health of caregivers in LMIC [20]. Young people with disabilities in LMIC are less likely to ever enroll school, have limited access to both education and health services, and are at increased risk of isolation, social exclusion, experiencing physical violence, and sexual and emotional abuse [30]. However, much of the research in LMIC settings seems to have been done on convenience samples, often in a major city or on caregivers attending a cerebral palsy (CP) center or in others already receiving help. 

Studies of caregivers of patients with mental disorders like schizophrenia and bipolar disorder in Nepal have shown a high level of stress-, depression-, and anxiety-related problems [31,32]. A study from a hospital in the city of Dhahran found that caretakers of intellectually disabled children in the hospital had a high level of anxiety and depression [33]. There seems to be a lack of knowledge of the caregiver situation especially in rural areas. To our knowledge, no studies on the psychological distress of NDD children’s caregivers have been published from rural Nepal. We have chosen to focus on the situation in rural areas, where most of the population in Nepal lives and where children with disabilities often receive no or very limited help.

Nepal has a difficult geography and infrastructure, and accessibility to roads is limited in many parts of the country. Eighty percent of the population lives in rural areas, and 40% of women are illiterate. It is among the poorest countries in the world, with about a quarter of its population living below the poverty line [34]. Studies have shown that the prevalence of mental illness such as depression in Nepal is similar to the prevalence in the developed world [35,36]. 

Cultural understanding, attitudes, and concepts of both mental health and disability differ and have an effect on stress, coping [37], and help-seeking behavior [38], and can thereby lead to ineffective treatment. Clarke and colleagues found that Nepali mothers’ concepts of psychological distress were often attributed to family- and gender-related factors. The distress often developed in a context of perceived duty towards the family and limited autonomy for women, and the response to distress was often shaped by a fatalistic worldview [39]. A report by Human Rights Watch found that families with disabilities are stigmatized and there is a strong belief in Nepal that disability is due to sins in a past life, fate, and God’s will. They also found that these beliefs often prevent caregivers from accessing appropriate education or healthcare for their disabled child because of shame, stigma, and an inability to see the possible benefit [40]. 

Resources for both disability and mental healthcare in Nepal are very limited, and there is a major lack of trained manpower, service centers, and integration of mental health into primary healthcare [41,42,43]. 

### Aim of the Study

The aim of this study was to estimate the psychological distress of caregivers with children suffering from NDD in rural Nepal and to find the factors associated with psychological distress.

## 2. Methods and Material 

### 2.1. Participants and Procedure

Participants were 63 primary caregivers of children aged 2–12 years suffering from NDD in four districts of Nepal (Chitwan, Bardia, Baglung, and Sunsari), covering eastern, western, and middle parts of the country. The inclusion criteria were as follows: primary caregivers of children aged 2–12 years with NDD [44]. The primary caregiver was defined as the person responsible for the day-to-day decision-making and care of the child. If the primary caregiver was a sibling under 18 years old, the person who had custodial authority (overall responsibility for care and control) was selected (see Table 1 for details).

Inclusion in the study was decided by trained child habilitation workers from the Self-Help Group for Cerebral Palsy (SGCP), an organization working with child habilitation in rural areas through a task-shifting approach. As there is no public register of children with NDD and these children were not enrolled for NDD treatment in the SGCP, information about possible children with NDD in the area was gathered from local people, youth clubs, local authorities, and other organizations. These children were then visited by a child habilitation worker, examined for possible NDD, and the caregiver invited to participate in the study. To our knowledge, none of the caregivers declined to participate in the study.

Data were collected through a semistructured interview administered by experienced, Nepalese female mental health personnel (a mental health nurse and a clinical psychologist) with broad experience of working with rural mental health projects. The interview guide was developed closely together with local expertise from Self-Help Group for Cerebral Palsy (SGCP) and Center for Mental Health and Counseling (CMC), to ensure the validity and relevance of the questions both culturally and linguistically. The interview guide was translated to Nepali, and the translation was discussed and then blind-translated back to English. 

The interviewers were trained in administering the questionnaire, and both the tools and the questionnaire were piloted on families with NDD in Kathmandu. All questions were read out loud and filled in by the interviewers, with clarification and explanation provided where necessary to ensure that the caregivers understood the questions. To secure the privacy and quality of the interviews, at least one of the team members entertained the child, other family members, and bystanders so the interviewer could be alone with the respondent. 

### 2.2. Measures

#### 2.2.1. Demographics

The caregivers reported their own age, their child’s age, marriage status, family type, literacy, education level, work outside the home, and partner’s working status.

#### 2.2.2. Diagnostic Assessment of NDD

The children were assessed for NDD by trained personnel from the SGCP on the basis of information from the caregivers and clinical observation prior to inclusion. During the subsequent interview, the children were again observed by these personnel, together with the research team, and all were confirmed to have NDD. 

Information on level of function and disability in the child was captured using the Ten Questions Screen [45], a disability screening tool especially developed for LMIC settings and previously used in Nepal [46]. 

For assessing the severity of motor disability, an adapted version of the *Gross Motor* Function Classification System (GMFCS) was used [47,48]. The GMFCS classifies the motor function of children from Level 1 to Level 5 with increasing severity. In addition, the caregiver was asked if the child was having difficulties with incontinence, feeding problems, or physical pain.

#### 2.2.3. Psychological Distress

Level of caregiver’s psychological distress was assessed using the Nepali version of the General Health Questionnaire-12 (GHQ-12) [49]. The GHQ-12 is a common assessment tool for mental well-being [50,51,52,53] and measures common mental health problems, including the domains of depression, anxiety, somatic symptoms, and social withdrawal. It has been found to be a sensitive instrument for detection of anxiety and depression in caregivers [54]. Patel and colleagues found the GHQ-12 to have the best balance of discriminating ability and internal consistency to identify cases of common mental disorders, but recommend using a higher cutoff (≥6) in very resource-limited settings [55]. 

The GHQ-12 includes 12 items, each with a four-point Likert scale, and asks questions like “Have you recently felt you couldn’t overcome your difficulties? (0) Not at all, (1) no more than usual, (2) rather more than usual, (3) much more than usual”. We used the binary scoring method recommended by Goldberg (1988) where the first two alternatives are coded 0 and the last two are coded 1, giving a maximum score of 12. As an indication of probable psychiatric “caseness”, a sum score ≥2 is recommended as a cutoff [56]. The Nepali validation of the GHQ-12 established a cutoff of ≥2, giving the scale good psychometric propriety with a sensitivity of 85.6% and a specificity of 75%, a positive predictive value of 87%, and a negative predictive value of 84% [49]. Cronbach’s alpha for the GHQ-12 in the present study was 0.787.

#### 2.2.4. Assessment of Impact of the Disabled Child on Caregiver’s Situation

The impact of the disabled child on the caregiver and family was measured with seven questions regarding the impact of the disabled child on central aspects of the caregiver’s life and situation, namely, household economy, caregiver’s physical health, workload, social life, marital relationship, dreams and expectations for their future, and effects on other siblings. We asked questions like “How has your child’s problems had an impact on your own physical health?” The questions were selected and formulated based on discussion with experienced health workers from Self-Help Group for Cerebral Palsy (SHGCP) and Center for Mental Health and Counseling (CMC). The answers were coded 0 (no negative impact), 1 (negative impact), or 2 (very negative impact). The scores on each of the seven items were summed, which gave a total score from 0 to 14, where a higher score indicates a higher burden on the caregiver. Cronbach’s Alpha for the caregiver impact score was 0.822. 

#### 2.2.5. Assessment of the Child’s Interaction and Participation in the Local Society 

Information about the child’s interaction and participation in society was obtained by five questions regarding the degree of social interaction compared to other children of the same age in the community, asking if the child had normal (coded 3), somewhat less (coded 2), or much less (coded 1) participation in the following settings: family, contact with people outside the family, school, children’s activities in the community, and attendance at activities and events in the local community. The assessments were done by questions like “When you compare your child to other children of his/hers age how much interaction (contact) does your child have with other people outside the family?”. These domains and formulations were selected on the basis of discussion with experienced local health workers from SHGCP and CMC. The scores on each of the five items were summed, which gave a total score of 5–15, where a low score indicates a low degree of interaction and participation in the local community. Cronbach’s Alpha for the child’s interaction and participation score was 0.829. 

### 2.3. Administration and Ethical Approval

Ethical approval was given by the Nepal Health Research Council and the Regional Committee for Medical and Health Research Ethics (REC West: 2012/2241) in Norway.

### 2.4. Statistics

Descriptive statistics (means, standard deviations, and percentages) were used to characterize the sample. To explore the relationship between child and caregiver characteristics and psychological distress (GHQ-12), we used Pearson’s correlations for continuous variables and Student’s *t*-test for dichotomous variables. In order to explore which of the factors explained the most variance of psychological distress, we also conducted a multiple forward linear regression, entering the GHQ-12 score as dependent variable and the significant factors from the correlational analysis as independent variables.

## 3. Results

### 3.1. Caregiver and Child Characteristics

A total of 63 primary caregivers were interviewed, all of them women: 60 (95%) were biological mothers, two (3%) were grandmothers, and one (2%) had another relation to the child (aunt). The majority of the caregivers (98%) were married. The age of the child ranged from 2 to 12 years (*M* = 6.66, *SD* = 2.73). Forty-one percent of the children were female. A summary of child and caregiver characteristics is given in Table 1. 

### 3.2. The Child’s Disability and Impact on the Caregiver

The mean GMFCS score was 3.60 (*SD* = 1.27). Most of the children had difficulties with weakness and stiffness (95.2%), movement (88.9%), unclear speech (73.0%), and ability to learn to do things like other children at the same age (73.8%). Sixty percent of the children were incontinent, and thirty-eight percent had difficulties with feeding. The answers to the Ten Questions Screen tool for disability are given in Table 2.

### 3.3. Impact of Disabled Child on Caregiver Burden (Caregiver Impact Score)

A majority of the caregivers experienced the child as having a negative or very negative impact on household economy (69.8%), their own physical health (65.1%), other siblings (44.4%), their workload (74.6%), their own social life (63.5%), and their own dreams and expectations for the future (81.0%); see Table 3 for details.

### 3.4. Psychological Distress 

Psychological distress among caregivers was measured using the GHQ-12. The mean GHQ-12 score was 5.38 (*SD* = 2.76, range = 0–12). A majority of 90.5% scored above the suggested cutoff (≥2) for screening for psychiatric cases in the community in Nepal [49], and 46% scored ≥6. The GHQ-12 scores are shown in Table 4.

There was no relationship between caregiver’s age, family type, or occupation on the GHQ-12 score. Caregivers who were illiterate had higher psychological distress (*M =* 6.24, *SD* = 3.15) compared with those who were literate (*M =* 4.82, *SD* = 2.34): *t* (61) = −2.06, *p* = 0.044. None of the ten disability questions were related to scores on the GHQ-12, and neither was the gender or age of the child. There was a significant relationship between the GHQ-12 score and the degree of disability (GMFCS: *r* = 0.418, *p* < 0.001) and feeding problems (*M* = 6.38, *SD* = 2.57) compared to those without feeding problems (*M* = 4.77, *SD* = 2.72): *t* (61) = 2.32, *p* = 0.023.

There was a significant relationship between all of the subscales on the child impact scale and the GHQ-12 score. Those who received help from health workers had significantly lower scores on the GHQ-12 (*M =* 4.43, *SD* = 2.02) compared with those who did not receive help (*M =* 5.93, *SD* = 2.99): *t* (59.29) = −2.35, *p* = 0.022. There was no significant relationship between any of the other help sources and the GHQ-12 score, including the total number of help sources. Those who had someone to confide in (87.3%) had significantly lower scores on the GHQ-12 (*M =* 5.09, *SD* = 2.54) compared with those who did not (*M =* 7.38, *SD* = 3.50): *t* (61) = −2.262, *p* = 0.027 (See Table 1 for details).

There was no significant relationship between the level of the child’s interaction with family or local community and the GHQ-12 score. The correlations between the GHQ-12, GMFCS, interaction, and impact scores are given in Table 5.

A multiple regression analysis, entering the significant factors from the correlational analyses, found that perceived caregiver burden (*β* = 0.57, *p* < 0.001), having a close friend to confide in (*β* = −0.23, *p* = 0.021), and having help from health workers (*β* = −0.21, *p* = 0.039) were significantly associated with GHQ-12. In total, the model explained 42.4% of the variance in psychological distress. See Table 6 for a summary of the multiple regression analysis. 

## 4. Discussion

In the current study, 90.5% of the caregivers scored above the suggested cutoff (≥2) on the GHQ-12. This is much higher than 11.7% found in the general female population in a rural district of Nepal [49]. Almost half of the caregivers (46%) scored six or higher, which is the optimum cutoff found by Patel and colleagues to use in very resource-poor settings to best balance between sensitivity and positive predictive value [55]. In contrast, about 10 % of mothers in a large study in a rural district of Nepal scored six or higher [39]. This indicates a higher psychological distress in caregivers of NDD children compared with what was previously reported in the general Nepali rural female population. The finding of a high level of distress in caregivers of NDD children is in line with previous studies of caregivers of children with intellectual disability [33] in Nepal and other studies of children with disabilities in an LMIC setting like India [28], Bangladesh [25], and Sri Lanka [29].

The correlation between child impact on caregiver, the GMFCS score, and psychological distress measured by the GHQ-12 demonstrates that the child’s disability seems to put a large overall burden on the caregivers and their families. This is in line with research that has shown that there is a considerable and neglected burden on caregivers in LMIC’s in terms of physical, psychological, social, time, and financial burden [3], and that the extra care burden related to having a disabled child can have a negative impact also on healthy siblings and marriage [23].

Previous findings have indicated that interventions and preventive strategies targeting caregivers may enable the caregivers to be more responsive to the child’s needs, thus decreasing the impact of the child’s disability on the caregivers [57], which can improve the wellbeing and the situation of the caregivers [58]. It is reasonable to expect similar results in a low-income setting. It is therefore important to address the situation of the whole family and not just the disabled child. Holistic interventions addressing the total family burden and the caregiver’s psychological health therefore seem important.

We found that psychological distress was strongly related to the severity of the child’s motor disability as measured by the GMFCS and to perceived caregiver impact. Some of this might be caused by increased workload related to having a severely disabled child. Interventions that can reduce workload may be useful to reduce caregiver’s stress. Lack of mobility aids, often combined with inaccessible roads, can be a challenge resulting in both the disabled child and the caregiver becoming housebound. Simple, affordable mobility aids have been used with success in several developing countries [59]. 

Likewise, feeding problems were associated with psychological distress in the caregiver, and a study from Bangladesh has shown that an intervention to improve the child’s feeding skills reduced maternal stress [60].

There was no correlation between the number of help sources and psychological distress. The only help source that was significantly correlated with lower psychological distress was advice and help from health workers, indicating that health workers might have an important role in reducing stress in caregivers. Having a close friend or relative to confide in was also significantly correlated to lower psychological distress as measured by the GHQ-12. This is in line with a review of Ribeiro (2013) that found that social support reduces stress levels in CP families [61]. Increasing social support might also therefore be important in order to improve the situation of these families. Community-based parent support groups have provided informal peer support as well as help and advice from health personnel [60,62].

Although one could assume that living in an extended family would provide more help for the caregivers, there was no association between psychological distress and family type. This has also been found in studies from Bangladesh [25] and Sri Lanka [29]. The finding of a high stress level in the present study can also be related to a strong feeling of duty towards the family [39], combined with a belief that disability is the result of sinful past actions [63] and the fate of god [40]. In addition, the whole family, but especially mothers of disabled children, is often stigmatized [40]. This might also hamper help-seeking behavior both for the disabled child and for the caregiver. 

Having a disabled child can be an economic challenge to the family because the caregiver often becomes housebound and has less possibilities for work outside the home [2,19]. In a setting with little or no governmental support, this may be an even greater problem, leading to increased poverty and debt [20]. The loss of income together with added expenses can have a detrimental effect on caregivers and their families in LMIC [3]. The correlation between disability and poverty is strong: poverty leads to higher rates of disability, and disability increases the risk of poverty [8]. 

Seventy percent reported that the child’s disability had either negative (32%) or very negative impact (38%) on their economy, indicating a substantial economic burden of having a disabled child. 

The finding that illiterate caregivers have higher psychological distress is in line with a study from Nigeria that showed that the most important predictors of a caregiver’s stress were severity of disability and level of education [64]. Brehaut and colleagues found that caregivers of disabled children had lower academic attainment, limited work opportunities, were more likely to be unemployed, and had lower income levels [17].

The forward regression analysis entering the significant factors from the correlational analysis indicated that caregiver burden, having a friend to confide in, and receiving help from health workers predicted psychological distress (GHQ12), after controlling for the effect of the other significant correlations. The factor that explained most of the variance was caregiver burden. This may indicate that measures to decrease caregiver burden are central to reduce psychological distress, as well as providing both informal psychosocial support and professional help. More studies are needed to understand the factors associated with caregiver’s psychological distress, as well as how to best introduce measures to reduce psychological distress among NDD caregivers in LMIC countries and in different cultural settings. 

### 4.1. Strengths and Limitations

In Nepal, 80% of the population reside in rural areas. Caregivers living here may have different challenges to caregivers living in a major city. It is therefore a strength that this study was performed in a rural setting and mainly among families who were not already enrolled in a program for NDD. Although case findings, infrastructure, and transportation can be a major challenge, generating knowledge about childhood disability and the caregiver’s situation in rural areas in LMIC is important in order to be able to develop better services for this group.

The use of experienced mental health professionals as interviewers and ensuring privacy in the interview setting are strengths of this study. Obtaining valid answers to personal questions, including questions regarding mental health issues, in a context where this is highly stigmatized and little spoken about, requires the interviewers to quickly build rapport and trust with the interviewee and be able to probe further when necessary.

The study has a number of limitations. The sample size is small, and statistical interpretation should be made with caution. Even though the sample was collected from different districts, Nepal is a very ethnically and culturally diverse country and the results may not be representative of the situation for NDD caregivers in rural Nepal. As there is no public register of children with NDD, information about possible children with NDD in the area was gathered from local people, youth clubs, local authorities, and other organizations, and cases might have been missed out. No participants refused to participate in the study. However some families may not have wanted to disclose that they have a disabled child. This may be a bias as such children would not be included in the study. 

### 4.2. Implications and Recommendations

The caregiver’s situation is often difficult, with a high burden of caregiving severely affecting mental health and quality of life. Lack of services for both disability and mental health seems to be adding to the burden of these caregivers. For most of the caregivers, there seemed to be no available mental health services locally. We suggest that interventions for this group will be more effective if they also involve a specialized and direct focus on caregiver’s mental health and coping. Sensitive care and support of the caregivers should be included in the follow-up of disabled children, and the caregivers could be screened with simple tools such as the GHQ-12 for more serious psychological distress. Decreasing caregiver burden and ensuring informal psychosocial support as well as support from health workers may be of importance to reduce psychological distress. 

## 5. Conclusions

The study found a high level of psychological distress and high overall burden in caregivers of children with NDD in four districts of Nepal. Perceived caregiver burden was associated with more psychological distress, while having a close friend to confide in and having help from health workers were associated with less psychological distress. More research is needed to understand and address the factor associated with psychological distress in NDD caregivers in the cultural and socioeconomic setting in which the families live their lives. 

## Figures and Tables

**Table 1 ijerph-18-02460-t001:** Caregiver and child characteristics and relationship to the General Health Questionnaire-12 (GHQ-12).

	Variable (Range/%)	*%*	M (*SD*)	*r*	*t*	*p*
**Caregiver characteristics**						
	Age in years (19–68)		31.51 (8.53)	0.056		0.662
	Nuclear family	50.8			1.269	0.209
	Illiteracy	39.7			2.059	0.044 *
	No schooling	49.2			1.211	0.231
	No school leaving certificate	90.5			0.353	0.725
	Not working outside home	76.2			0.183	0.856
	Migrant worker partner	28.1			0.259	0.797
**Help sources**			2.29 (1.16)	−0.176		0.166
	Help from partner	82.5			−0.216	0.830
	Help from grandparents	42.9			1.084	0.283
	Siblings	17.5			−0.743	0.460
	Other family members	30.2			−1.639	0.106
	Health workers	36.5			−2.354	0.022 *
	Help from others	19			−0.066	0.948
	Have somebody to confide in	87.3			−2.262	0.027 *
**Child characteristics**						
	Age in years (2–12)		6.66 (2.73)	−0.039		0.762
	Female ^a^	41			−1.128	0.264
	GMFCS score (1–5)		3.60 (1.27)	0.418		0.001 *
	Child’s interaction score (5–15)		9.45 (3.29)	−0.108		0.408
	Receiving incentive from government	46			3.019	0.004 *

Note: * significant at *p* < 0.05; *r* = Pearson correlation coefficient, ^a^ = female coded 1 and male 0. GMFCS: Gross Motor Function Classification System.

**Table 2 ijerph-18-02460-t002:** Ten Questions Screen tool for disability and relationship to the GHQ-12.

Child Experience Significant Problem with:	Yes (%)	*t/r*	*p*
1. Delays in motor development	98.4	−0.589	0.558
2. Reduced vision	12.7	0.130	0.897
3. Reduced Hearing	17.7	0.025	0.980
4. Understanding simple commands	13.1	0.761	0.450
5. Weakness/stiffness	95.2	0.672	0.504
Movement	88.9	0.385	0.701
6. Fits/epilepsy	32.3	1.635	0.107
7. Learning	73.8	0.860	0.393
8. Speaking at all	50.8	1.55	0.125
9.Unclear speech	73.0	0.049	0.961
10. Mental slowness	90.5	0.821	0.415
**Sum (Range 5–10; *M* = 5.58; SD = 1.21)**		0.121 ^a^	0.360
Other problems			
Incontinence	60.3	0.794	0.430
Feeding	38.1	2.324	0.023 *
Pain	87.3	1.678	0.098

Note: * significant at *p* < 0.05, ^a^ Pearson correlation coefficient.

**Table 3 ijerph-18-02460-t003:** Caregiver impact score.

Negative Impact of Child’s Disability on:	Impact (%)			Mean (SD)	Relationship to GHQ-12	
	No Negative Impact	Negative Impact	Very Negative Impact		*r*	*p*
Household economy	30.2	31.7	38.1	1.08 (0.8)	0.45	<0.001 **
Your physical health	34.9	42.9	22.2	0.87 (0.8)	0.44	<0.001 **
Your workload	25.4	34.9	39.7	1.14 (0.8)	0.48	<0.001 *
Other siblings	55.6	33.3	11.1	0.56 (0.7)	0.36	0.004 *
Your social life	36.5	34.9	28.6	0.92 (0.8)	0.30	0.016 *
Your marital relationship	87.3	4.8	7.9	0.21 (0.6)	0.28	0.028 *
Your own dreams and expectations for the future	19	49.3	31.7	1.13 (0.7)	0.49	<0.001 **
Sum score (range 0–14)				5.91 (3.61)	0.579	<0.001 **

Note: * significant at *p* < 0.05; ** significant at *p* < 0.01; *r* = Pearson correlation coefficient.

**Table 4 ijerph-18-02460-t004:** The GHQ-12 scores (*n* = 63).

GHQ-12 Questions	Binary Scoring (0-0-1-1)	
	0 No	Yes
	%	%
(1) Lost much sleep over worry?	49.2%	50.8%
(2) Felt constantly under strain?	34.9%	65.1%
(3) Not been able to concentrate?	68.3%	31.7%
(4) Not felt that you are playing a useful part in things?	77.8%	22.2%
(5) Not been able to face up to your problems?	76.2%	23.8%
(6) Not felt capable of making decisions about things?	63.5%	36.5%
(7) Felt you couldn’t overcome your difficulties?	42.9%	57.1%
(8) Not been feeling reasonably happy, all things considered?	36.5%	63.5%
(9) Not been able to enjoy normal day-to-day activities?	63.5%	36.5%
(10) Been feeling unhappy or depressed?	28.6%	71.4%
(11) Been losing confidence in yourself?	49.2%	50.8%
(12) Been thinking of yourself as a worthless person?	71.4%	28.6%

**Table 5 ijerph-18-02460-t005:** Pearson correlations between GHQ-12, GMFCS, interaction, and impact scores.

	GMFCS	Child Interaction	Child Impact
GHQ-12	0.418 **	−0.108	0.579 **
GMFCS		−0.651 **	0.418 *
Interaction			−0.235

Note: GMFCS = Gross Motor Function Classification System score; child interaction = child’s interaction score; child impact = child’s impact score. Correlation is significant at: * *p* < 0.05 and ** *p* < 0.01 (two-tailed).

**Table 6 ijerph-18-02460-t006:** Forward regression analysis for factors associated with psychological distress (GHQ-12).

		*B*	*β*	*p*
Step 1	Constant	2.767		<0.001
	Caregiver burden	0.442	0.579	<0.001
Step 2	Constant	4.736		<0.001
	Caregiver burden	0.440	0.576	<0.001
	Friend or relative to confide in	−2.244	−0.273	0.008
Step 3	Constant	4.916		<0.001
	Caregiver burden	0.435	0.569	<0.001
	Friend or relative to confide in	−1.927	−0.234	0.021
	Help from health workers	−1.204	−0.209	0.039

## Data Availability

The data presented in this study are available on request from the corresponding author. The data are not publicly available due to privacy restrictions.

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
