# Peer review of "Psychological Distress among Caregivers of Children with Neurodevelopmental Disorders in Nepal"

_ijerph, 2021, doi:10.3390/ijerph18052460_

Round 1

Reviewer 1 Report

Dear authors, 

I revised your paper. Although the theoretical background is well organized and the overall purpose is relevant, I think the current form of the paper did not extend the knowledge in the field of the interplay between parental stress and child pathological condition. The only correlation between two variables is not sufficient. I suggest to revise the statistic strategies consistently. 

Author Response

Response to Reviewer 1 Comments

Point 1: I revised your paper. Although the theoretical background is well organized and the overall purpose is relevant, I think the current form of the paper did not extend the knowledge in the field of the interplay between parental stress and child pathological condition. The only correlation between two variables is not sufficient. I suggest to revise the statistic strategies consistently. 

Response 1:

Thank you for your response regarding the background section and overall purpose of the article. We agree that analyses that are more complex should be used in order to examine the factors that could explain psychological distress among caregivers having children with NDD. We have therefore extended the statistics with regression analysis, and included the findings in the discussion and conclusion.

All line references to the manuscript are to “Simple Markup view mode” in word.

“A multiple regression analysis entering the significant factors from the correlational analyses found that perceived caregiver burden (β =0.57., p<0.001), having a close friend to confide in (β = -0.23, p=0.021), and having help from health workers (β = -0.21, p=0.039) were significantly associated with GHQ-12. See Table 6 for a summary of the multiple regression analysis. In total, the model explained 42.4 % of the variance in psychological distress (see Table 6 for details)“ (Line 310-315)

We have updated the abstract, implications and recommendations and the conclusion sections based on these findings:

Abstract: “There was a significant relationship between the caregiver’s psychological distress (GHQ-12) and degree of disability in the child (Gross Motor Function Classification System), degree of caregiver burden, feeding problems, having health workers as a possible source of help, receiving incentive from the government, having somebody to confide in and caregiver illiteracy. A forward regression analysis entering the significant factors indicated that caregiver burden, having someone to confide in and having health workers as possible source of help was significant. The final step of the model explained 42.2% of the variance in psychological distress among the caregivers. The study indicates a high level of psychological distress and high overall burden in caregivers of children with NDD in rural Nepal.” (Line 24-31)

Discussion: “The forward regression analysis, entering the significant factors from the correlational analysis, indicated that caregiver burden, having a friend to confide in and receiving help from health workers predicted psychological distress (GHQ12), after controlling for the ef-fect of the other significant correlations. The factor that explained most of the variance were caregiver burden. This may indicate that measures to decrease caregiver burden is central to reducing psychological distress, as well as providing both informal psychoso-cial support and professional help. More studies are needed to both understand the factors associated with caregiver’s psychological distress, as well as how to best introduce measures to reduce psychological distress among NDD caregivers in LMIC countries and in different cultural settings.” (Line 402-411)

Implications and recommendations: “Decreasing caregiver burden and ensuring both informal psychosocial support as well as support from health workers may be of importance to reduce psychological distress ” (Line 445-446)

Conclusion: “Perceived caregiver burden was associated with more psychological distress, while having a close friend to confide in and having help from health workers were associated with less psychological distress. More research is needed to understand and address the factor associated with psychological distress in NDD caregivers in the cultural and socioeconomic setting the families live their lives.” (Line 449-454)

We agree that we could have been clearer on why the article could add to the current knowledge base on caregivers of NDD children’s situation and have therefore included the following background information in the introduction:

“Children with disabilities are often not represented in official statistics and as a result remain politically and socially ‘invisible’ (UNICEF, 2020).” (Line 48-49)

“There is a growing understanding of the impact of NDD on the quality of life and mental health of families with children with these conditions. However, the majority of research has been conducted in high-income countries and there is a lack of knowledge on the situation in low and middle income countries (Scherer et al., 2019).”  (Line 60-63)

We believe that the situation of caregivers of disabled children in rural areas of LMIC is an under-researched topic and that there is a clear need to gain more knowledge. According to Thrush and Hyder (2014), the lack of information, especially quantitative data, on the situation of caregivers and their burden in LMIC allows the perpetuation of the misleading notion that families in LMIC are capable of providing care without public or private institutional support (Thrush and Hyder, 2014) and that better evidence would provide incentive for governments of LMIC to put the needs of caregivers higher on their agenda. To our knowledge, this is the first article on NDD children’s caregivers' psychological distress from rural Nepal. 

Reviewer 2 Report

This paper is a report of a study conducted in Nepal on the psychological well-being of caregivers of children with neurodevelopmental disorders. Although the study has various limitations (which the authors are aware of), the major one being the small study sample, the results are very important. It addresses the problem identified in the introduction, indicating that in the absence of data, it can be assumed that families in low- and middle-income countries (such as Nepal), particularly in rural areas, are capable of providing care without public or private institutional support. The presented findings fill this gap and indicate that the problem of lack of support for caregivers of such children exists and needs closer investigation. Moreover, the results presented highlight the important role of health workers in reducing stress in caregivers.

The article has a clear structure and presents a coherent narrative. The study was well thought out, conducted and described - I have no methodological concerns here. It is worth noting the details such as preparing a professional translation into Nepali or involving an additional person who takes care of the interviewee's "surroundings" (family members and bystanders), so that the interviewee can focus on the interview without any distractions. The paper concludes with a critical evaluation of the findings and an attempt to draw recommendations for mental health management in such areas.

While reading the article, I found three minor errors:

  1. In Table 1 there is no information how sex is coded. I assume that Female is coded as 1, and Male as 0 as sex-related row is named "Female", but this is not explicitly stated. This is only needed to understand the associated t-value (which is not significant).
  2. On lines 203-204 it was indicated that the responses were "combined". In Table 3 one can read that it is a simple sum, but it is not obvious from the text in the paragraph itself - some more sophisticated method could have been used. I propose to clarify this.
  3. I also found simple technical errors. In the abstract on line 25 there is ";" instead of ":". In the subsequent line, in turn, there is a space missing after the period. Also, in Tables 3 and 5, the first element of the first column is unnecessarily bold.

I also have a discussion point. An interesting aspect of the problem, concerning the stigmatization of those suffering from disabilities, is presented in lines 113-120. I wonder if it would not be valuable to include additional parts in the interview concerning this issue in the follow-up study, to see if there are variations among such beliefs and how they affect well-being of caregivers.

To conclude, the paper is soundly prepared and addresses issues that are important for public health, so I recommend its acceptance to the IJERPH journal.

Reviewer 3 Report

The aim of this manuscript is to examine the psychological distress among caregivers having children with neurodevelopmental disorder (NDD) in rural Nepal. The article is clearly written and taps on an important and relatively under-researched topic. However, several points should be clarified and/or expanded and more information should be provided before the manuscript can be accepted for publication.

  1. Authors should use more updated references in the Introduction. For example, this figure is about 16 years ago: “It is estimated that 150 million children suffer from some kind of 40 disability, and most of these live in low-income countries (UNICEF, 2005).”
  2. “Overall, NDD conditions are likely to be more common in low- and middle-income countries (LMIC) because of the higher prevalence of recognized risk factors such as poor maternal and…”- this is a central sentence in the study but it has no reference.
  3. Page 2 line 54- “this research” however, authors provide more than one study
  4. In the Introduction authors should provide more information about all the factors they used. For example: “Assessment of the child’s interaction and participation in the local society.”
  5. For all questionnaires authors should include: Internal reliability and an item example.
  6. In the Measures section- authors should add the background information they collected.
  7. The statistics analyses are good, however, there is a need to add more complex analyses such as regressions or path analysis in order to examine the factors that explain psychological distress among caregivers having children with NDD.

Round 2

Reviewer 1 Report

0 words

Reviewer 3 Report

Thanks for the opportunity to review this paper. Authors addressed all comments and the paper is now suitable for publication.